# Tumor suppressor p73 induces apoptosis of murine peritoneal cell after exposure to hydatid cyst antigens; a possibly survival mechanism of cystic echinococcosis *in vivo* mice model

Ehsan Ahmadpour[1], Adel Spotin[2], Ata Moghimi[3], Firooz Shahrivar[1], Farhad Jadidi-Niaragh[3], Farnaz Hajizadeh[2], Sirous Mehrani[1], Komeil Mazhab-Jafari[4,5]*

1 Infectious and Tropical Diseases Research Center, Tabriz University of Medical Sciences, Tabriz, Iran, 2 Immunology Research Center, Tabriz University of Medical Sciences, Tabriz, Iran, 3 Drug Applied Research Center, Tabriz University of Medical Sciences, Tabriz, Iran, 4 Department of Laboratory Sciences, Abadan University of Medical Sciences, Abadan, Iran, 5 Department of Parasitology, School of Medicine, Ahvaz Jundishapur University of Medical Sciences, Ahvaz, Iran

* k.mjafari@yahoo.com

## Abstract

Cystic echinococcosis (CE) is a life-threatening helminthic disease caused by the *Echinococcus granulosus* sensulato complex. Previous evidence indicates that the host's innate immune responses against CE can combat and regulate the growth rate and mortality of hydatid cyst in the host's internal organs. However, the survival mechanisms of CE are not yet fully elucidated in the human body. In the present study, the apoptotic effects of fertile and infertile hydatid fluid (HF) were tested on murine peritoneal cells *in vivo* mice model. Mice were divided into five groups including; control group, fertile HF-treated peritoneal cells, infertile HF-treated peritoneal cells, protoscolices (PSCs)-treated peritoneal cells and HF+PSCs-treated peritoneal cells group. Mice groups were intraperitoneally inoculated with PBS, HF, and/or PSCs. Afterwards, peritoneal cells were isolated and mRNA expression of STAT3, caspase-3, p73 and Smac genes were evaluated by quantitative Real-time PCR. After 48 hours of exposure, the protein levels of Smac and STAT3 was determined by western blotting technique. After 6 hours of exposure, Caspase-3 activity was also measured by fluorometric assay. The intracellular reactive oxygen species (ROS) production was examined in all groups. The mRNA expression levels of p73, caspase-3 and also Caspase-3 activity in HF+PSCs-treated peritoneal cells were higher than in the test and control groups ($Pv < 0.05$), while the mRNA expression level of anti-apoptotic STAT3 and Smac genes in HF+PSC-treated peritoneal cells were lower than in the other groups ($Pv < 0.05$). As well, the level of intracellular ROS in the fertile HCF-treated peritoneal cells, infertile HCF-treated peritoneal cells, PSC-treated peritoneal cells and HF+PSC-treated peritoneal cells groups were significantly higher than in the control group ($Pv < 0.05$). Current findings indicates that oxidative stress and p73 can trigger the apoptosis of murine peritoneal cells through

**Data Availability Statement:** All relevant data are within the paper and its Supporting information files.

**Funding:** The research project has been funded and approved by the code "98u-265" by the Abadan University of Medical Sciences, Abadan, Iran. The funders had no role in study design, data collection and analysis, decision to publish, or preparation of the manuscript.

**Competing interests:** The authors have declared that no competing interests exist.

modulator of HF-treated PSCs that is likely one of the hydatid cyst survival mechanisms *in vivo* mice model.

---

# 1. Introduction

*Echinococcus granulosus* (*E. granulosus*) is one of the well-known species of the cestoda that causes cystic echinococcosis (CE)/hydatidosis. Its distribution in the world depends on environmental and anthropogenic factors [1,2]. The prevalence of CE is about 1-200/100,000 in the world, according to the official data, the most affected areas are Mediterranean countries, Russia and China [3,4]. The life cycle of this parasite is such that the canids plays the definitive host role, and human as an accidental dead-end host is infected due to consumption of contaminated vegetables by egg, direct contact with canids and putting contaminated hands in the mouth [5,6]. *Echinococcus* eggs contain an embryo called an oncosphere or six-hooked embryo [7]. The main target organs of hydatid cysts are in the liver parts (most hepatic cysts are located in the right lobe), lung and other unusual places, such as bone, brain, spleen, kidney, etc [8,9]. Recent immunological findings have revealed new aspects about the innate immune responses (apoptosis, inflammasome and toll-like receptors) generated during the establishment of CE; however, many escaping mechanisms of parasite in the hydatid cyst-host cross-talk have not been fully investigated *in vivo* model yet [10].

Aforementioned mechanism, apoptosis, referred to as programmed cell death, is a regulated form of cell death that involves distinct biochemical and morphological changes. Apoptosis has different pathways and can be activated intrinsically and extrinsically, both of these pathways end with caspase-3 activation [11]. In the intrinsic pathway, which is called the mitochondria pathway, different types of proteins play reciprocate roles in inducing and down regulating of apoptosis. For example, Bax proteins, p53 family and caspase family have auxiliary roles and STAT3, Bcl-2, and Smac have inhibitory roles. These molecules affect each other in different ways and can have positive and negative regulatory effects on the function [12–14]. Apoptosis of *E. granulosus* protoscolices (PSCs) was first identified more than a decade ago by a group of researchers from Chile [15]. The p53 family of transcription factors comprises three proteins: p53, p63 and p73 [16]. According to some evidence, we hypothesis that tumor suppressor p73 possibly induces apoptosis by activation of intrinsic (mitochondrial) apoptotic pathway [17]. However, the mechanism of p53-dependent apoptosis has not fully yet been understood in HF-treated peritoneal cells. In the present study, the oxidative stress and apoptotic effects of fertile and infertile hydatid fluid (HF) were tested on murine peritoneal cells *in vivo* mice model.

# 2. Materials and methods

## 2.1. Mice and ethics statement

In this study, 25 healthy non-infected female mice, aged 6–8 weeks, were purchased from Biology Supply Center at Pasteur research Institute (Tehran, Iran). Mice were housed inside cages with free access to the standard pellet animal diet and water. The study was approved by the Scientific Research Ethical Committee, Abadan University of Medical Sciences (IR.ABADA-NUMS.REC.1398.030). Also, animal handling and all procedures were done in accordance with the international ethical guidelines.

## 2.2 Preparation of PSCs and hydatid layer

The hydatid liver cysts of sheep were collected from a slaughterhouse in Abadan and transferred to the parasitology laboratory of Abadan University of Medical Sciences. The contents

of the cysts were aspirated by a sterile syringe, shed in a sterile glass cylinder, and settled for 2–3 minutes. Next, the supernatant was discarded, and the precipitate was washed three times with phosphate buffer (pH 7.2). The viability of PSCs was evaluated by using trypan blue stain under an optical microscope. After 5 min, dead PSCs were blue color, while the living ones were colorless. The suspension, with at least 90% of PSCs being alive, was transferred to a dark container and kept at 4°C.

## 2.3 Immunization and challenge

Mice were intraperitoneally administrated by hydatid fluid (HF) and PSCs using previously described method [18,19]. In short, mice were divided into five groups (each group number; five mice) including; control group (saline normal), fertile HCF-treated peritoneal cell, infertile HCF-treated peritoneal cell, PSCs-treated peritoneal cell and HF+PSCs-treated peritoneal cell group. Subsequently, animals were injected intraperitoneally with either 0.2 mL of hydatid fluid (2000 PSCs) in experimental and saline for control groups. At 48 hours post-infection, the mice were euthanized with an overdose of ketamine/xylazine, subsequently peritoneal cells were isolated by centrifugation at 2000 $g$ for 15 min at 4°C.

## 2.4. RNA extraction, cDNA synthesis, and quantitative real-time PCR

To evaluate the mRNA expression levels of STAT-3, p73, caspase-3 and Smac genes, isolated peritoneal cells were subjected to RNA isolation using TRIzol solution (YTA, Tehran, Iran) [20]. The quantity and quality of the isolated RNA was then assessed by a Nanodrop 2000c spectrophotometer (Thermo Scientific, USA). The process of cDNA synthesis was performed using cDNA Synthesis Kit (YTA, Tehran, Iran), according to the manufacturer's instructions.

Subsequently, relative mRNA expression levels of STAT3, p73, caspase-3 and Smac genes in peritoneal cells were evaluated by quantitative real-time-PCR (qRT-PCR) technique. qRT-PCR was carried out in a final volume of 20 μL containing 0.2 μM of each primer set (Table 1), 9 μL of SYBR green reagent (YTA, Iran), 1 μL of cDNA template and 8 μL of nuclease-free water. The PCR reaction was carried out by an initial denaturation step at 95° C for 3 minutes, and the 45 cycles of 95° C for 10 seconds, 58° C for 30 seconds and 72° C for 20 seconds (Livak and Schmittgen 2001). Glyceraldehyde-3-phosphate dehydrogenase (GAPDH) expression levels were considered as an internal control (housekeeping gene) for data normalization.

**Table 1. Set of specific gene primers used in Real-time PCR.**

| Gene | Primer | Product length | Accession Number |
|---|---|---|---|
| P73 | F:5′ -AGAGCATGTGACCGACATTGTT-3′ | 103 bp | XM_006538722.5 |
| | R: 5′ -TTCTACACGGATGAGGTGGCT-3′ | | |
| Smac/Diablo | F: 5′ -AGGAGGAAGATGAGGTGTGG -3′ | 248/127 bp | XM_036165411.1 |
| | R: 5′ -TCAGCAGCCATCTCTGAAAG-3′ | | |
| STAT3 | F: 5′ -GGGCCATCCTAAGCACAAAG-3′ | 113 bp | XM_011248846.4 |
| | R: 5′ - GGTCTTGCCACTGATGTCCTT-3′ | | |
| Caspase-3 | F: 5′- TGTCATCTCGCTCTGGTACG - 3′ | 201 bp | XM_017312543.3 |
| | R: 5′- AAATGACCCCTTCATCACCA- 3′ | | |
| GAPDH | F:5′ - CCTCGTCCCGTAGACAAAA - 3′ | 102 bp | NG_148718.1 |
| | R: 5′ - AATCTCCACTTTGCCACTG - 3′ | | |

Abbreviations: R, common reverse primer; F. forward primer; bp, base pair; RT PCR, real time PCR,.

## 2.5. Immunoblotting analysis

To evaluate the protein levels of STAT3 and Smac, mouse peritoneal cells were subjected to western blot analysis. For this case, cells were homogenized in a 500 μL RIPA buffer containing protease inhibitor cocktail by a homogenizer after incubation for 30 min at 4°C, followed by centrifugation at $12000 \times g$ for 10 min. subsequently protein concentrations were quantified using Bradford method. Before running samples in 10% SDS polyacrilamide gel, protein samples were boiled at 95°C for 5 minutes and 20 μg of protein/ sample was loaded in each well. Afterwards, protein bands were transferred to PVDF membrane and blocked using 1% milk/TBS for 1 hour at room temperature. Desired proteins were probed in 1% milk/TBS supplemented by specific primary antibodies of STAT3 (sc-8019, 1:300), Smac (sc-136071, 1:300) and β-actin (sc-47778, 1:300) over night at 4°C. Next day, after several washing steps, specific horseradish peroxidase (HRP) conjugated secondary antibody were probed (sc-516102, 1:1000 for STAT3 and Smac and sc-2357, 1:1000 for β-actin) and incubated 2 hour at room temperature. The protein bands were visualized using a chemiluminescence detection kit (ECL, Bio-Rad) and radiographic film (Kodak, USA). The density of bands was measured by ImageJ software. β-actin was used as a loading control.

## 2.6. DCFH-DA assay

The level of ROS was quantified using the fluorescent probe $2'$, $7'$- dichlorofluorescein-diacetate (DCFH-DA), according to the method of Driver *et al.*, with a slight modification [21]. Homogenates from each group were diluted in ice-cold's buffer to obtain a concentration of 5 mg tissue/mL. The homogenates were pipetted into 96-well plates and allowed to warm up to room temperature for 5 min. 10μM of DCFH-DA was added to each well, and the plates were incubated for 15 min at room temperature to allow the DCFH-DA to be incorporated into any membrane-bound vesicles. The converted DCF product was measured using the multiple readers with a fluorescence spectrophotometer, excitation at 485 nm and emission at 530 nm. The measured fluorescence values were expressed as a percentage of the fluorescence with respect to those of the exercised control.

## 2.7. Detection of Caspase-3 activity

Caspase-3 activity was determined by the Caspase-3/CPP32 Fluorimetry Assay Kit (K105) (BioVision Inc., Mountain View, CA, USA) as manufacturer's manual. The assay was based on detection of cleavage of substrate DEVD-AFCA (AFC: 7-amino-4-trifluoromethyl coumarin). Activity was measured by a Fluorimetry (Jasco, FP-6200, Japan), as enzyme activity converts a blue emission (λmax = 400 nm) to a yellow-green color (λ = 505nm), which can be measured using a fluorimetry or a fluorescence microtiter plate reader.

## 2.8. Statistical analysis

Data analyzes were carried out by GraphPad Prism version 8.0.1 Software (GraphPad Software Inc., La Jolla, CA, USA). The analysis of variance (ANOVA) and the independent t test were used to compare the differences between the study groups.

## 3. Results

The results of Caspase-3/CPP32 fluorometric assay indicated that the Caspase-3 activity in HF +PSCs-treated peritoneal cells was significantly higher than that of fertile HF-treated peritoneal cell ($Pv<0.05$), infertile HF-treated peritoneal cell, PSCs-treated peritoneal cell and

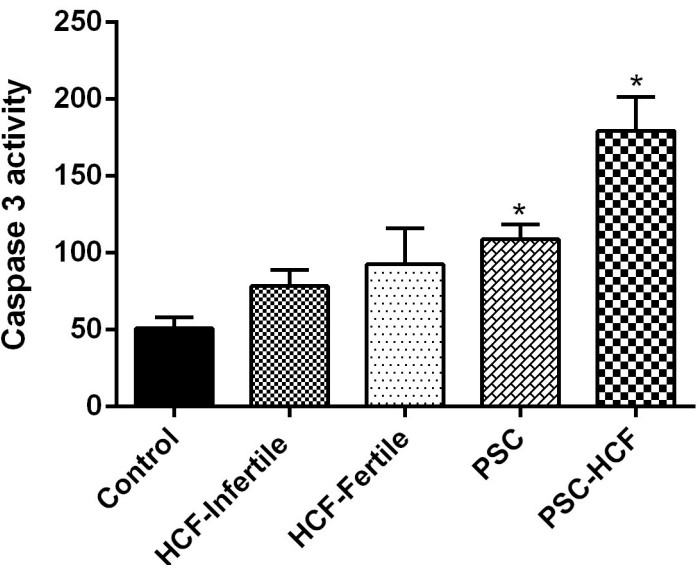

**Fig 1. Evaluation of Caspase-3 activity in cell extracts.** Caspase-3 activity in fertile HF, infertile HF, PSCs, PSCs+HF and cell control (from left to right). Bar graph indicates the mean±SEM. Increase in Caspase-3 activity was determined by comparing fluorescence of 7-amino-4-trifluoromethyl coumarin in control.

control group, respectively (Fig 1). Based on the results of the qRT-PCR assay, significant changes were observed in the expression of apoptotic genes compared with control group.

The highest mRNA expression levels of p73 and caspase-3 genes were observed in the peritoneal cell exposed to PSCs+HF, PSCs, fertile HF, infertile HF groups compared with control group, respectively ($Pv < 0.05$) (Fig 2A and 2B). While the mRNA expression levels of STAT3 and Smac genes in the HF+PSCs-treated peritoneal cell group was significantly lower than that in the control groups ($Pv < 0.05$) (Fig 2B and 2C).

Western blot analysis indicated that the expression of Smac protein was significantly decreased in the PSCs+HF, PSCs, fertile HF groups compared to infertile HF and control group (Fig 3A and 3B) ($Pv < 0.05$), as well expression of STAT3 protein was significantly decreased in the PSCs+HF and PSCs groups compared to control group ($Pv < 0.05$) (Fig 3B).

As shown in Fig 4, level of intracellular ROS (the intensity of DCF fluorescent) in the PSCs +HF and PSCs groups ($Pv < 0.01$) were significantly higher than those in the control group (Fig 4B).

## 4. Discussion

*E. granulosus* has significant defense mechanisms against the host's immune response, which permits the survival of hydatid cyst in the host body [22]. Hydatid cyst can live in the human body for up to 53 years [23]. The identification of innate signaling pathways for *Echinococcus* persistence in the host target cells provides a promising way to implement the appropriate preventive strategies for hydatidosis [24]. However, little is known about innate immune responses between host and parasite cross-talk in CE patients *in vivo* model.

Apoptotic effects of HF antigens and PSCs have been previously studied on various cancer lines, macrophages and T lymphocytes, suggesting that hydatid cyst metabolites, particularly fertile cyst components, have the ability to induce apoptosis via both internal (mitochondrial)

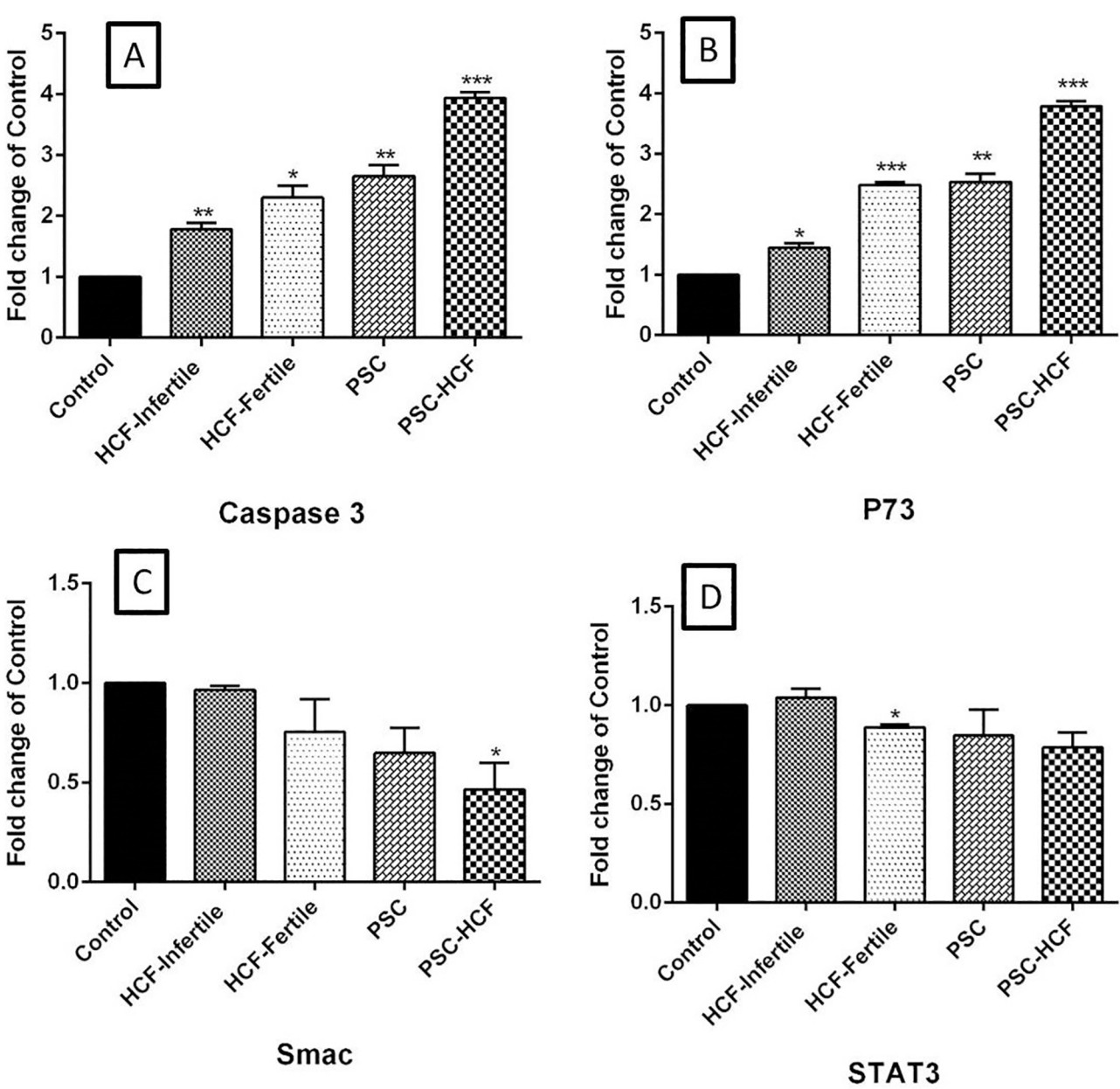

**Fig 2. Relative fold change of caspase-3, p73, Smac and STAT3 in hydatid fluid groups in comparison to the control group.** Results are expressed as mean and SD. P-value was determined using unpaired two-tailed t-test (*$Pv<0.05$, ** $Pv<0.01$, *** $Pv<0.001$, **** $Pv<0.0001$).

and external (death receptor) pathways [23,25]. Among the known signaling pathways, apoptotic bifunctional effects in the relationship between host's lymphocytes and PSCs of CE have already been reported by Spotin *et al.*, by assessing the suppression and survival mechanisms of the hydatid cyst [26].

Current results showed that apoptosis can be induced by activating of caspase-3 in murine peritoneal cells after exposure to HF-treated PSCs through tumor suppressor p73 and ROS. It has been previously verified that hydatid fluid toxins have lytic effects on mouse peritoneal macrophages [23].

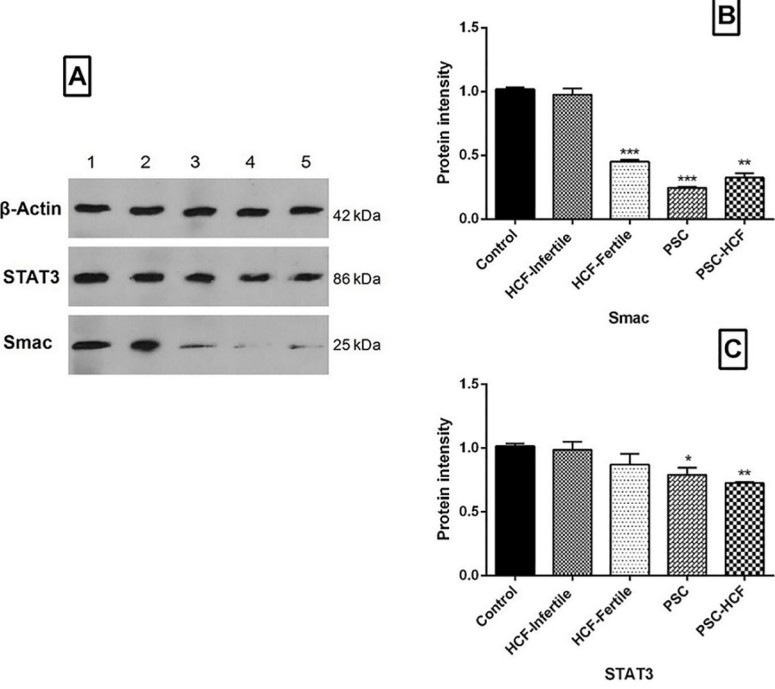

**Fig 3. Immunoblotting of STAT3 and Smac in peritoneal cell among different groups (A), Quantitation of Immunoblotting of Smac (B) and STAT3 (C).**

This study shows that both Caspase-3 activity and caspase-3 mRNA expression were significantly higher in fertile HF-treated peritoneal cells compared to infertile HF-treated peritoneal cells and control cells, while the expression level of STAT3 and Smac genes as anti-apoptotic molecules was reduced in the fertile HF treated peritoneal cells relative to the infertile HF-

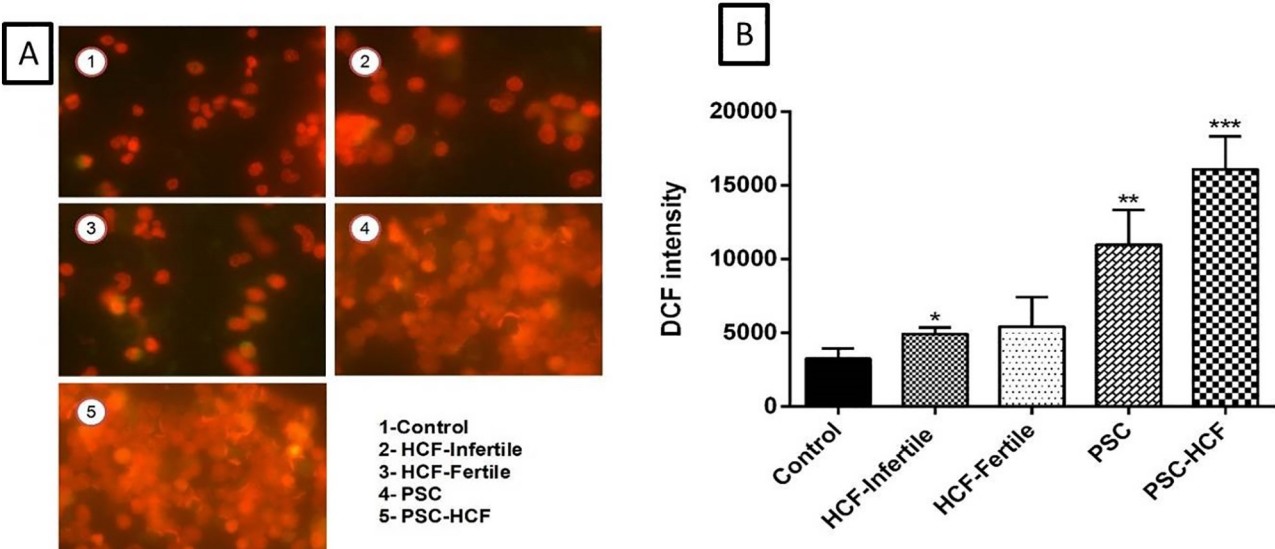

**Fig 4. Effects of inflammation on the peritoneal cells reactive oxygen species (ROS) production.** Fluorescent microscopic images of PI & DCF-stained (Left) Quantified PI & DCF fluorescence (Right).

treated peritoneal cells and control. Similarly, Mokhtari Amirmajdi *et al.*, showed that the Caspase-3 activity and the ratio of Bax/Bcl-2 mRNA expression were higher in PSCs-treated lymphocytes relative to infertile fluid-treated lymphocytes and control group which exhibited apoptosis could be as a possible mechanism by which *E. granulosus* overwhelms host defenses [23]. Interestingly, Spotin *et al.*, showed that mRNA expression of apoptosis-inducing ligands (TRAIL and Fas-L) increases in germinal layer of infertile cysts compared to fertile cyst and healthy tissue [27].

Analysis of DNA fragmentation and caspase-3 activity in germinal layer showed higher levels of apoptosis in infertile cysts compared to fertile cysts, suggesting that apoptosis is a possible mechanism of infertility in hydatid cysts [15]. Depending on the cellular pathway, the p53 tumor suppressor interferes with multiple pathways, including cell cycle arrest, cellular senescence, and triggering of apoptosis dependent to the intrinsic pathway [28,29]. On the other hand, the role of the p53 gene family in the CE patients and parasite relationship has not yet fully understood. The results of the current study indicate that p73 can trigger the Caspase-3 activity of murine peritoneal cells through modulator of HF-treated PSCs by downregulating of Smac and STAT3 genes.

A similar study showed that p53 level increases in human lung adenocarcinoma (A549) cell lines exposed to hydatid cyst fluid compared to human healthy lung epithelial (BEAS-2B) but hydatid cyst fluid did not directly cause cell death [30]. Likewise, the tumor suppressor p53 induces apoptosis of host's lymphocytes experimentally infected by *Leishmania major*, by activation of Bax and caspase-3 [17].

In this study, the intracellular ROS level in the HF+PSCs-treated peritoneal cells were significantly higher than in the control group, indicating that ROS likely induces peritoneal cell apoptosis through mitochondrial damage, ASK1 activation, and/or PARP activation to escape the host's immune responses [31]. Based on recent studies, the potential role of TLR2/TL4 polymorphism has been presented as a predisposing factor in patients with recurrent hydatidosis (RH) [32,33]. Moradkhani *et al.*, (2019) have shown that the homozygous mutant-type TLR2 Gln/Gln (A/A) is associated with the occurrence of RH and conferred a 9 fold risk for susceptibility [32]. Noori *et al.*, (2018) have also demonstrated that the heterozygous mutant-type TLR4 Asp299Gly genotype has a tendency to be associated with the occurrence of RH and conferred a 3-fold risk for susceptibility [33].

## 5. Conclusions

In conclusion, the oxidative stress and tumor suppressor p73 induces apoptosis of host's peritoneal cells experimentally infected with HF and/or PSCs, by activation of Caspase-3, which can potentially be addressed as an ancillary role of the survival mechanism of hydatid cyst in CE patients. These results can enhance our knowledge on the design and implementation of anti-apoptotic treatments that can inhibit the growth of hydatid cyst in infected patients. Apoptosis of murine peritoneal cells through modulator of HF-treated PSCs should be a possible survival mechanism in CE *in vivo* model.

## Supporting information

**S1 Fig. This is the original Western blot image of B- actine.**
(JPG)

**S2 Fig. This is the original Western blot image of Smac.**
(JPG)

**S3 Fig. This is the original Western blot image of STAT3.**
(JPG)

## Author Contributions

**Conceptualization:** Ehsan Ahmadpour, Adel Spotin, Firooz Shahrivar, Komeil Mazhab-Jafari.

**Data curation:** Ehsan Ahmadpour, Adel Spotin, Firooz Shahrivar, Komeil Mazhab-Jafari.

**Formal analysis:** Ehsan Ahmadpour, Adel Spotin, Firooz Shahrivar, Farhad Jadidi-Niaragh, Sirous Mehrani, Komeil Mazhab-Jafari.

**Funding acquisition:** Ehsan Ahmadpour, Komeil Mazhab-Jafari.

**Investigation:** Ehsan Ahmadpour, Adel Spotin, Ata Moghimi, Farhad Jadidi-Niaragh, Farnaz Hajizadeh, Sirous Mehrani, Komeil Mazhab-Jafari.

**Methodology:** Ehsan Ahmadpour, Adel Spotin, Firooz Shahrivar, Farhad Jadidi-Niaragh, Farnaz Hajizadeh, Komeil Mazhab-Jafari.

**Project administration:** Ehsan Ahmadpour, Adel Spotin, Komeil Mazhab-Jafari.

**Resources:** Ehsan Ahmadpour, Adel Spotin, Farhad Jadidi-Niaragh, Komeil Mazhab-Jafari.

**Software:** Ehsan Ahmadpour, Adel Spotin, Farhad Jadidi-Niaragh, Farnaz Hajizadeh, Sirous Mehrani, Komeil Mazhab-Jafari.

**Supervision:** Ehsan Ahmadpour, Adel Spotin, Komeil Mazhab-Jafari.

**Validation:** Ehsan Ahmadpour, Adel Spotin, Sirous Mehrani, Komeil Mazhab-Jafari.

**Visualization:** Ehsan Ahmadpour, Adel Spotin, Sirous Mehrani, Komeil Mazhab-Jafari.

**Writing – original draft:** Ehsan Ahmadpour, Adel Spotin, Ata Moghimi, Firooz Shahrivar, Farhad Jadidi-Niaragh, Sirous Mehrani, Komeil Mazhab-Jafari.

**Writing – review & editing:** Adel Spotin, Ata Moghimi, Firooz Shahrivar, Farhad Jadidi-Niaragh, Sirous Mehrani, Komeil Mazhab-Jafari.

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
