## [Decision Letter · Decision Letter 0]

26 Jun 2023

PONE-D-23-17969Tumor suppressor p73 induces apoptosis of murine peritoneal cell after exposure to hydatid cyst antigens; a possibly survival mechanism of cystic echinococcosis in vivo mice modelPLOS ONE

Dear Dr. Mazhab-jafari,

Thank you for submitting your manuscript to PLOS ONE. After careful consideration, we feel that it has merit but does not fully meet PLOS ONE’s publication criteria as it currently stands. Therefore, we invite you to submit a revised version of the manuscript that addresses the points raised during the review process.

We look forward to receiving your revised manuscript.

Kind regards,

Muhammad Mazhar Ayaz, Ph.D

Academic Editor

PLOS ONE

Journal Requirements:

https://link.springer.com/article/10.1007/s00436-017-5517-8

https://www.sciencedirect.com/science/article/abs/pii/S0001706X21003806?via%3Dihub

In your revision ensure you cite all your sources (including your own works), and quote or rephrase any duplicated text outside the methods section. Further consideration is dependent on these concerns being addressed.

4. We suggest you thoroughly copyedit your manuscript for language usage, spelling, and grammar. If you do not know anyone who can help you do this, you may wish to consider employing a professional scientific editing service.  

5. Please note that funding information should not appear in any section or other areas of your manuscript. We will only publish funding information present in the Funding Statement section of the online submission form. Please remove any funding-related text from the manuscript.

  "Funding and ethics approval: The research project has been funded and approved by the code "98u-265" and with the ethics code "IR.ABADANUMS.REC.1398.030" at the Abadan University of Medical Sciences, Abadan, Iran."

7. Please amend the manuscript submission data (via Edit Submission) to include author Farnaz Hajizadeh.

8. Your ethics statement should only appear in the Methods section of your manuscript. If your ethics statement is written in any section besides the Methods, please delete it from any other section. 

9. PLOS ONE now requires that authors provide the original uncropped and unadjusted images underlying all blot or gel results reported in a submission’s figures or Supporting Information files. This policy and the journal’s other requirements for blot/gel reporting and figure preparation are described in detail at https://journals.plos.org/plosone/s/figures#loc-blot-and-gel-reporting-requirements and https://journals.plos.org/plosone/s/figures#loc-preparing-figures-from-image-files. When you submit your revised manuscript, please ensure that your figures adhere fully to these guidelines and provide the original underlying images for all blot or gel data reported in your submission. See the following link for instructions on providing the original image data: https://journals.plos.org/plosone/s/figures#loc-original-images-for-blots-and-gels. 

Reviewers' comments:

Reviewer's Responses to Questions

**Comments to the Author**

1. Is the manuscript technically sound, and do the data support the conclusions?

Reviewer #1: Yes

Reviewer #2: Yes

2. Has the statistical analysis been performed appropriately and rigorously? 

Reviewer #1: I Don't Know

Reviewer #2: Yes

3. Have the authors made all data underlying the findings in their manuscript fully available?

Reviewer #1: Yes

Reviewer #2: Yes

4. Is the manuscript presented in an intelligible fashion and written in standard English?

Reviewer #1: No

Reviewer #2: Yes

5. Review Comments to the Author

Reviewer #1: • The sections " RNA extraction and cDNA synthesis" and "Real-time PCR" could be merged and presented in one section.

• In table 1 please mention the accession number of each gene.

• In the sentence "After the end of the blocking time, the paper is incubated with the primary antibody mixed and diluted with the blocking solution to a certain amount of β-actin primary antibody (sc-47778, 1:300) for 16 to 18 hours." please mention the specification of all primary antibody which used in this study?

• In the section "Detection of Caspase-3 activity" please mention the method of total protein extraction. Furthermore, How did the authors evaluate the concentration of extracted proteins?

• Figure 1 should be edited the y-axis label. I think the relative caspase 3 activity was proper.

Reviewer #2: This manuscript entitled “Tumor suppressor p73 induces apoptosis of murine peritoneal cell after exposure to hydatid cyst antigens model'' presents a possibly survival mechanism of cystic echinococcosis in vivo mice model. Importantly, this MS highlights oxidative stress and p73can trigger the apoptosis of murine peritoneal cells through modulator of HF-treated PSCs that is likely one of the hydatid cyst survival mechanisms in vivo mice. As a result, it is a well-organized MS. However, before any proceeding some improvements should be addressed into new version.

Comments

Some parts of the article should be revised in terms of English and grammar.

Abstract:

The term “in vivo” should be “italic” entire paper

It seems that “Some paraphrasing” should be done. E.g. line 41: “Mice groups were intraperitoneally inoculated with HF and PSC”

Line 53: “PSC stathis likely” corrected to “PSCs that is likely

Keywords: the “Hydatidosis” changed to “cystic echinococcosis”

“Introduction”

Line 60” the “causescystic” corrected to “causes cystic”

Line 81” After repeating echinococcus once, it should be abbreviated to “ E. granulosus “ in the rest of article.

-Head and sub-heads of MS should be numbered.

- Results and discussion should be written separately.

In addition to apoptosis, it is suggested that in the discussion section, the role of innate immunity of TLR 2 and 4 in the survival of the Hydatid cyst in the human body be investigated based recent published articles:

1- https://link.springer.com/article/10.1007/s00436-018-5850-6

2- https://www.sciencedirect.com/science/article/abs/pii/S014795711930133X

6. PLOS authors have the option to publish the peer review history of their article (what does this mean?). If published, this will include your full peer review and any attached files.

Reviewer #1: **Yes: **Saeid Afshar

Reviewer #2: No

---

## [Author Response · Author response to Decision Letter 0]

14 Aug 2023

Answers to comments of Reviewers

We would like to thank you very much for the very thoughtful review that helped us greatly to revise the manuscript and to improve its quality.

Comments to the Author

1. Is the manuscript technically sound, and do the data support the conclusions?

Reviewer #1: Yes

Reviewer #2: Yes

2. Has the statistical analysis been performed appropriately and rigorously? 

Reviewer #1: I Don't Know

Reviewer #2: Yes

3. Have the authors made all data underlying the findings in their manuscript fully available?

Reviewer #1: Yes

Reviewer #2: Yes

4. Is the manuscript presented in an intelligible fashion and written in standard English?

Reviewer #1: No

Reviewer #2: Yes

Answer: Thank you for your attention. The manuscript was checked again for grammar and spelling.

5. Review Comments to the Author

Reviewer #1: 

• The sections "RNA extraction and cDNA synthesis" and "Real-time PCR" could be merged and presented in one section.

Answer: We thank the honorable reviewer for their valuable input and suggestion. Sections were merged based on the reviewer's comment. 

• In Table 1 please mention the accession number of each gene.

Answer: Thank you for your attention. The accession numbers were added. 

• In the sentence "After the end of the blocking time, the paper is incubated with the primary antibody mixed and diluted with the blocking solution to a certain amount of β-actin primary antibody (sc-47778, 1:300) for 16 to 18 hours." please mention the specification of all primary antibody which used in this study?

Answer: Thank you for your suggestion. This section was revised and it was added.

• In the section "Detection of Caspase-3 activity" please mention the method of total protein extraction. Furthermore, How did the authors evaluate the concentration of extracted proteins?

Answer: For detection of Caspase-3 activity, about 10 peritoneal cells were re-suspended with 50 μl of chilled Cell Lysis Buffer (provided in the kit) and incubated on ice for 10 minute. Then the protein extracted from these cells was evaluated according the kit instruction. 

• Figure 1 should be edited the y-axis label. I think the relative caspase 3 activity was proper.

Answer: Thank you for your attention. It was edited. 

Reviewer #2:

 This manuscript entitled “Tumor suppressor p73 induces apoptosis of murine peritoneal cell after exposure to hydatid cyst antigens model'' presents a possibly survival mechanism of cystic echinococcosis in vivo mice model. Importantly, this MS highlights oxidative stress and p73can trigger the apoptosis of murine peritoneal cells through modulator of HF-treated PSCs that is likely one of the hydatid cyst survival mechanisms in vivo mice. As a result, it is a well-organized MS. However, before any proceeding some improvements should be addressed into new version.

Comments

Some parts of the article should be revised in terms of English and grammar.

Abstract:

The term “in vivo” should be “italic” entire paper

Answer: It was done and highlighted into new version.

It seems that “Some paraphrasing” should be done. E.g. line 41: “Mice groups were intraperitoneally inoculated with HF and PSC”

Answer: It was done and highlighted into new version.

Line 53: “PSC stathis likely” corrected to “PSCs that is likely

Answer: It was done and highlighted into new version.

Keywords: the “Hydatidosis” changed to “cystic echinococcosis”

Answer: It was done and highlighted into new version.

“Introduction”

Line 60” the “causescystic” corrected to “causes cystic”

Answer: It was done and highlighted into new version.

Line 81” After repeating echinococcus once, it should be abbreviated to “ E. granulosus “ in the rest of article.

Answer: It was done and highlighted into new version.

-Head and sub-heads of MS should be numbered.

Answer: It was done.

- Results and discussion should be written separately.

Answer: It was done.

In addition to apoptosis, it is suggested that in the discussion section, the role of innate immunity of TLR 2 and 4 in the survival of the Hydatid cyst in the human body be investigated based recent published articles:

1-https://link.springer.com/article/10.1007/s00436-018-5850-6

2- https://www.sciencedirect.com/science/article/abs/pii/S014795711930133X

Answer: It was done and highlighted into new version.

6. PLOS authors have the option to publish the peer review history of their article (what does this mean?). If published, this will include your full peer review and any attached files.

Do you want your identity to be public for this peer review? For information about this choice, including consent withdrawal, please see our Privacy Policy.

Reviewer #1: Yes: SaeidAfshar

Reviewer #2: No

---

## [Editor Report · Decision Letter 1]

21 Sep 2023

Tumor suppressor p73 induces apoptosis of murine peritoneal cell after exposure to hydatid cyst antigens; a possibly survival mechanism of cystic echinococcosis in vivo mice model

PONE-D-23-17969R1

Dear Dr. Mazhab-jafari,

We’re pleased to inform you that your manuscript has been judged scientifically suitable for publication and will be formally accepted for publication once it meets all outstanding technical requirements.

Kind regards,

Asif Ali, PhD

Academic Editor

PLOS ONE
---

## [Editor Report · Acceptance letter]

25 Sep 2023

PONE-D-23-17969R1 

Tumor suppressor p73 induces apoptosis of murine peritoneal cell after exposure to hydatid cyst antigens; a possibly survival mechanism of cystic echinococcosis *in vivo* mice model 

Dear Dr. Mazhab-jafari:

I'm pleased to inform you that your manuscript has been deemed suitable for publication in PLOS ONE. Congratulations! Your manuscript is now with our production department. 

Kind regards, 

on behalf of

Dr. Asif Ali 

Academic Editor

PLOS ONE